# Chain of Preference Optimization:
# Improving Chain-of-Thought Reasoning in LLMs

**Xuan Zhang**[*12], **Chao Du**[†1]**, Tianyu Pang**[1]**, Qian Liu**[1]**, Wei Gao**[2]**, Min Lin**[1]
[1]Sea AI Lab, Singapore
[2]School of Computing and Information Systems, Singapore Management University
xuanzhang.2020@phdcs.smu.edu.sg; weigao@smu.edu.sg;
{duchao, liuqian, tianyupang, linmin}@sea.com

## Abstract

The recent development of chain-of-thought (CoT) decoding has enabled large language models (LLMs) to generate explicit logical reasoning paths for complex problem-solving. However, research indicates that these paths are not always deliberate and optimal. The tree-of-thought (ToT) method employs tree-searching to extensively explore the reasoning space and find better reasoning paths that CoT decoding might overlook. This deliberation, however, comes at the cost of significantly increased inference complexity. In this work, we demonstrate that fine-tuning LLMs leveraging the search tree constructed by ToT allows CoT to achieve similar or better performance, thereby avoiding the substantial inference burden. This is achieved through *Chain of Preference Optimization* (CPO), where LLMs are fine-tuned to align each step of the CoT reasoning paths with those of ToT using the inherent preference information in the tree-search process. Extensive experimental results show that CPO significantly improves LLM performance in solving a variety of complex problems, including question answering, fact verification, and arithmetic reasoning, demonstrating its effectiveness. Our code is available at https://github.com/sail-sg/CPO.

## 1 Introduction

Recent advances in large language models (LLMs) have shown that constructing reasoning chains is critical to improving their problem-solving capabilities [1, 2, 3, 4, 5, 6, 7]. A representative method is chain-of-thought (CoT) [1], which prompts LLMs to generate intermediate reasoning steps, i.e., thoughts, thereby constructing explicit reasoning paths (as depicted in Figure 1(a)). While straightforward and intuitive, recent research observes that CoT can often overlook optimal reasoning paths and exhibit an unconscious style of answering due to its single-path focus [8, 9]. To foster a more deliberate and conscious reasoning style, Yao et al. [8] propose tree-of-thought (ToT), which generates multiple branching thoughts at each step of the reasoning process and conducts self-evaluation for pruning and planning to search for reasoning paths (as shown in Figure 1(b)). However, despite improving reasoning quality, ToT significantly increases computational complexity, which limits its practical application. This raises the question: Can the strategic depth of ToT be integrated into CoT to enhance its effectiveness while maintaining efficiency?

Existing research has initially provided a positive answer to the above question [10, 11, 12]. A natural strategy is to treat the reasoning path discovered by ToT for each instance as a target for supervision, and then fine-tune LLMs to improve their CoT reasoning abilities [11, 12]. Several methods have been proposed to improve this approach, including using advanced tree-search techniques like Monte

---

*Work done during Xuan Zhang's associate membership at Sea AI Lab. †Correspondence to Chao Du.

38th Conference on Neural Information Processing Systems (NeurIPS 2024).

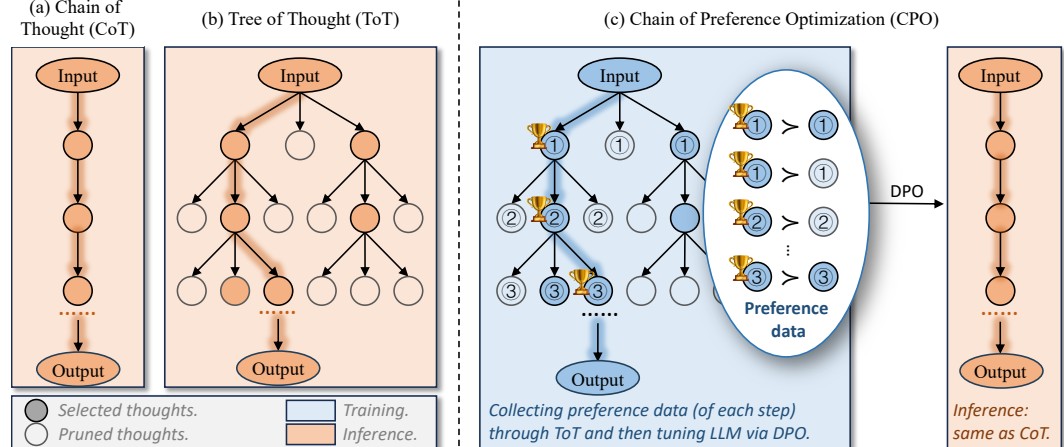

Figure 1: Comparison of CoT, ToT, and CPO methods, where each node illustrates a step in the reasoning process, forming coherent language sequences aimed at solving a problem. The highlighted path indicates the chosen reasoning trajectory. In the CoT method, the LLM generates only one new node at each step, and all generated nodes are used to build the final reasoning path. For ToT, the LLM produces $k$ new nodes at each step, but only the top n-best nodes are kept, with the rest being pruned. In CPO, nodes marked with a trophy represent preferred thoughts, while those marked with numbers are nodes that can be utilized to create preference data. This method uses the search tree structure from ToT to develop paired preference data, subsequently training LLMs to align with these preferences through DPO.

Carlo tree-search (MCTS) and employing external reward models [12, 10] for pruning and planning to gather better reasoning paths as supervision. The effectiveness of these approaches is therefore largely dependent on the quality of the best-discovered reasoning path.

In this paper, we identify a limitation in these approaches: they overlook the non-optimal reasoning thoughts generated during the tree-search process, which naturally provides additional preference information. Specifically, ToT inherently generates multiple alternative thoughts at each reasoning step, and pruning is performed according to their evaluated qualities. This tree-search process constitutes a preference over all *intermediate* thought candidates—thoughts appearing in the best-discovered reasoning path are preferred over those that do not. Moreover, this could shed even more insights than the final best-discovered reasoning path, as non-optimal reasoning paths (and thus preferences) exist at each step in the tree-search.

Inspired by recently developed reinforcement learning from human feedback (RLHF) techniques like direct preference optimization (DPO) [13], we propose *Chain-of-Preference Optimization* (CPO) to fully exploit the inherent preference information. Specifically, we construct paired preference thoughts at each reasoning step according to the search tree of ToT and then train LLMs to align with these preferences using the DPO algorithm (as illustrated in Figure 1(c)). The paired preference thoughts are constructed based on the above intuition: at each reasoning step, we categorize thoughts as preferred or dispreferred based on their inclusion in the final paths chosen by ToT. With such preference data, CPO enables LLMs to generate the path preferred by ToT using CoT decoding at inference time.

We conduct extensive experiments to evaluate the effectiveness of CPO. Experiments on seven datasets using LLaMA [14] and Mistral [15] as base models demonstrate that CPO is highly effective in teaching LLMs the preferred thoughts of ToT at each reasoning step, leading to an average accuracy improvement of up to $4.3\%$ compared to the base models. Additionally, the experiments reveal that CPO can achieve comparable or even superior performance to the ToT method, which on average requires more than $50$ times longer for inference.

## 2 Related Work

**Reasoning with LLMs.** LLMs have been shown to perform better when prompted to engage in multi-step reasoning [1, 2, 3]. Many studies have focused on improving the generated reasoning paths

by post-editing [16] or accessing external knowledge [3, 17]. A distinct approach, more relevant to our interests, transforms the linear reasoning structure into a non-linear format such as a tree or graph [18, 19, 20, 8, 9, 21], which combines thought evaluation with search algorithms like depth-first search (DFS) [22]. Different from our proposed CPO, these methods require searching during inference, which significantly increases latency.

**LLM self-improving.** Reinforcement learning (RL) has increasingly been applied to LLMs by treating them as RL agents for alignment with human feedback [23, 24, 25, 26]. Recent advances demonstrate the potential of using LLMs for self-generating data to augment fine-tuning processes [27, 28, 29, 30, 31, 32]. For instance, reinforced self-training methods [33, 34, 35, 36, 37] introduce mechanisms to curate new high-quality examples and iteratively enrich the training dataset for enhancing model performance. Nevertheless, these methods typically rely on either an external reward model [35, 34] or labeled data [33]. In contrast, approaches like self-rewarding [29, 38] utilize LLMs themselves to evaluate the generated content, aligning more closely with our method. However, these strategies still require initial seed data [29, 38], necessitating human annotation. Our work differs from previous methods as it does not rely on any ground-truth data, allowing LLMs to self-learn from their own feedback. Additionally, our approach constructs feedback in a chain fashion, focusing on reasoning steps, an aspect overlooked by prior works.

**Monte Carlo tree-search for LLMs.** Monte Carlo tree-search (MCTS) is a robust algorithm for navigating complex decision-making environments, commonly employed in strategic board games such as AlphaGo [39, 40, 41, 42, 43]. MCTS methodically constructs a search tree, balancing exploration and exploitation, simulates various outcomes, and updates utility estimates based on these simulations. Recent studies have shown that MCTS can enhance the decoding process in LLMs [11, 44, 45, 21, 12]. However, the primary challenge with MCTS is the high latency during inference, particularly in difficult reasoning tasks [46, 47]. While some approaches have attempted to optimize LLMs by leveraging reasoning paths identified through MCTS [11, 12], these methods still rely on labeled data and require separate policy and value models to explore and evaluate potential moves at the tree's leaves. In contrast, our CPO approach eliminates the need for human annotations and simplifies the tuning of LLMs without the necessity for additional models.

## 3 Background

In this section, we formalize our notation and provide a brief overview of key prior knowledge for our method. We denote language sequences by lowercase letters, e.g., $x$, $y$, $z$, to represent a sequence of tokens. The output distribution of an LLM parameterized by $\theta$ is denoted by $\pi_\theta$.

### 3.1 Chain-of-Thought Prompting

Chain-of-thought (CoT) [1] is a method that prompts LLMs to generate a chain of reasoning steps before the final answer, as shown in Figure 1. It introduces a series of intermediate thoughts, denoted as $z_1, \cdots, z_n$, that link an input $x$ to an output $y$, where $n$ is the total number of reasoning steps. For instance, if $x$ is a combination of demonstration examples and the input question and $y$ is the final answer, each intermediate thought $z_i$ forms a coherent language sequence representing a part of the overall reasoning path toward the final answer. The demonstration examples consist of a set of CoT demonstrations, which serve as exemplars in the prompting process. The intermediate reasoning thoughts are sequentially sampled from the distribution $z_i \sim \pi_\theta(\cdot|x, z_1, \cdots, z_{i-1})$ and the output is then derived from $y \sim \pi_\theta(\cdot|x, z_1, \cdots, z_n)$.

### 3.2 Tree-of-Thought Prompting

Tree-of-thought (ToT) [8] enables LLMs to explore multiple reasoning paths before answering a given question, as illustrated in Figure 1. This approach models the LLM reasoning task as a search over a tree, where each node represents a thought step in the reasoning path. ToT comprises two main components, both implemented through prompting LLMs: 1) the *thought generator* and 2) the *state evaluator*. The *thought generator* constructs several new thoughts for the next step based on the current state. Subsequently, the *state evaluator* generates scores for each new thought and selects the

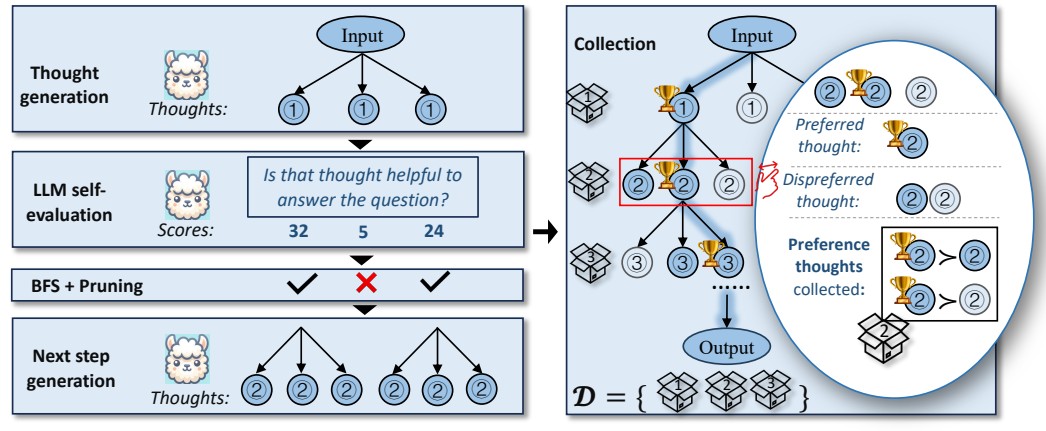

Figure 2: The framework of our CPO method. The left part illustrates the process of generating, evaluating, and pruning thoughts, while the right part demonstrates the collection of preference thoughts. The shaded path represents the final selected reasoning path. Thoughts marked with a trophy indicate preferred data, while sibling nodes without a trophy are marked as dispreferred.

n-best thoughts for further search. The final result is determined by the search algorithm (e.g., BFS or DFS) applied over the selected thoughts until the reasoning process reaches a conclusion.

## 3.3 Direct Preference Optimization

Direct preference optimization (DPO) is a method for directly optimizing an LLM to align with preference data [13], e.g., human feedback [13, 48, 49]. More specifically, RLHF traditionally frames the application of human feedback to enhance the performance of an LLM within the context of an RL problem. However, DPO reformulates the reward modeling and RL fine-tuning phases in RLHF to a single optimization problem. The objective function of DPO aims to maximize the ratio of probabilities for the preferred responses and optimize the LLM to imitate human preferences.

Given the generations $(\hat{y}_1, \hat{y}_2) \sim \pi(\hat{y}|x)$ conditioned on input $x$, these pairs are evaluated and ranked according to specific criteria. Preference data is then constructed from these ranked pairs, denoted by $\hat{y}_w \succ \hat{y}_l | x$, where $\hat{y}_w$ and $\hat{y}_l$ denote the preferred (winning) and dispreferred (losing) completions between $\hat{y}_1$ and $\hat{y}_2$, respectively. The DPO objective is formulated as follows:

$$\mathcal{L}_{\text{DPO}}(\pi_\theta; \pi_{\text{ref}}) = -\log \sigma \left( \beta \log \frac{\pi_\theta(\hat{y}_w|x)}{\pi_{\text{ref}}(\hat{y}_w|x)} - \beta \log \frac{\pi_\theta(\hat{y}_l|x)}{\pi_{\text{ref}}(\hat{y}_l|x)} \right), \tag{1}$$

where $\sigma$ is the logistic function, the hyperparameter $\beta$ regulates the penalty imposed for the deviations from the base reference model $\pi_{\text{ref}}$.

## 4 Our Method: Chain of Preference Optimization

Unlike previous methods that train LLMs to learn the complete reasoning paths [10, 50, 11, 12], our approach leverages the preferences over thoughts generated at each reasoning step, which are often discarded in prior works. Our key insight is that non-optimal thoughts generated during the tree-search process in ToT provide valuable preference information that can enhance LLM's reasoning ability. A major advantage of our method is that it utilizes this supervision only during training, thereby avoiding high inference latency. Our approach consists of two components: synthesizing the chain of preference thoughts (i.e., the preference thoughts in a chain fashion) and training with the CPO objective.

### 4.1 Synthesizing the Chain of Preference Thoughts

Our procedure for synthesizing and collecting preference thought pairs closely follows the inference process of ToT [8]. An overview of our method is shown in Figure 2. Specifically, the detailed process is divided into three parts: 1) *thought generation*, which generates multiple thoughts for each

reasoning step; 2) *state evaluation*, which evaluates each thought; and 3) *search and collection*, which finalizes the preference thoughts.

**Thought generation.** Given a state $s_{i-1} = [x, z_1, \cdots, z_{i-1}]$ representing a partial solution with the input $x$ and the sequence of thoughts $[z_1, \cdots, z_{i-1}]$ so far, we sample $k$ thoughts for the next reasoning step:

$$z_i^{(j)} \sim \pi_\theta(z_i|s_{i-1}) = \pi_\theta(z_i|x, z_1, \cdots, z_{i-1}), \quad \text{for } j = 1, \cdots, k. \tag{2}$$

Conditioned on the initial input $x$, which contains the demonstration examples and the question to be answered, and the previous thoughts $z_1, z_2, \cdots, z_{i-1}$, the LLM generates multiple thoughts for the next reasoning step. Specifically, it follows the format of demonstrations, starting with the prefix "`Step` $i$," and samples $k$ thoughts $\{z_i^{(j)}\}_{j=1}^k$. We control the model to pause at the end of $z_i^{(j)}$ by setting the generation of the string "`Step` $i+1$," as the stop criteria.[2] As a result, we obtain $k$ new states $s_i^{(j)} = [x, z_1, \cdots, z_{i-1}, z_i^{(j)}]$ for $j = 1, \cdots, k$.

**State evaluation.** Given different states $\{s_i^{(j)}\}_{j=1}^k$, we utilize the LLM to reason about the states and evaluate their progress toward solving the problem, eliminating the need for an external reward model or human annotations. To evaluate state $s_i^{(j)}$, the input to the LLM includes specific demonstration examples for the evaluation process, the input question $x$, and all the thoughts in the state (i.e., $[z_1, \cdots, z_{i-1}, z_i^{(j)}]$). The LLM follows the format of demonstrations to generate a verbal justification first, followed by a classification result from two classes: `likely` and `impossible`. The classification results are then used to assign a score, with `likely` $= 10$ and `impossible` $= 1$.

The prompt template used in our evaluation consists of two parts: (1) the general guidelines, and (2) task-specific demonstration examples. To minimize the effects of randomness and bias, we shuffle the order of demonstration examples [51] and repeatedly sample the generated justification and evaluation results. We then calculate the average score for the state $s_i^{(j)}$. The general guideline prompt for the evaluation is as follows: `Evaluate whether the thought helps in partially or directly answering the original question (likely/impossible).`

**Search and collection.** We use BFS with pruning as the search algorithm to select the reasoning paths. After evaluation, we retain the n-best thoughts with the highest evaluation scores and proceed to the next step of generation. When the LLM generates a thought containing "`so the final answer is:`", the search algorithm concludes and returns the selected paths.

As shown in the right part of Figure 2, after finalizing the reasoning paths, the thoughts within the selected paths are marked as preferred (i.e., winning) thoughts. For each preferred thought at the $i$-th step $z_i^w$, we construct corresponding dispreferred (i.e., losing) thoughts. First, we identify the parent state $s_{i-1}^w$, which includes all the previous thoughts leading to $z_i^w$. Each child thought of $s_{i-1}^w$ that is not included in the selected path is chosen as a dispreferred thought $z_i^l$ compared to $z_i^w$. This process results in the preference pair $(z_i^w, z_i^l)$ for the state $s_{i-1}^w$. We highlight that the constructed dataset $\mathcal{D}$ includes *preference data at every step of the reasoning chain*. This per-step paired preference supervision is usually overlooked in previous methods [11, 12].

### 4.2 Training with the CPO Objective

Once we have obtained the chain of preference thoughts $\mathcal{D}$, we can proceed with optimization. For the $i$-th step, given the previous reasoning thoughts $s_{i-1}^w$, the probabilities of generating $z_i^w$ and $z_i^l$ are denoted as $\pi_\theta(z_i^w|x, s_{i-1}^w)$ and $\pi_\theta(z_i^l|x, s_{i-1}^w)$, respectively. To optimize the LLM on this pair of preference thoughts, we can directly substitute it into Equation 1:

$$\mathcal{L}_i(\pi_\theta; \pi_{\text{ref}}) = -\log \sigma \left( \beta \log \frac{\pi_\theta(z_i^w|x, s_{i-1}^w)}{\pi_{\text{ref}}(z_i^w|x, s_{i-1}^w)} - \beta \log \frac{\pi_\theta(z_i^l|x, s_{i-1}^w)}{\pi_{\text{ref}}(z_i^l|x, s_{i-1}^w)} \right). \tag{3}$$

---

[2]The "stop criteria" is used to control when generation should stop, which is implemented via a function input in Hugging Face's Transformers Library.

Thus, the objective function for CPO can be formulated as follows:

$$\mathcal{L}_{\text{CPO}}(\pi_\theta; \pi_{\text{ref}}) = \mathbb{E}_{(x, z_i^w, z_i^l, s_{i-1}^w) \sim \mathcal{D}} \left[ \mathcal{L}_i(\pi_\theta; \pi_{\text{ref}}) \right]. \tag{4}$$

## 5 Experiments

In this section, we empirically validate that CPO improves the reasoning ability of the base model, and uncover several insightful findings.

### 5.1 Settings

**Datasets and evaluation metrics.** We focus our research on three types of reasoning tasks: *Question Answering* (QA), *Fact Verification*, and *Arithmetic Reasoning*. For QA, we conduct experiments on three widely used datasets: Bamboogle [17], WikiMultiHopQA [52], and HotpotQA [53]. For fact verification, we use three datasets: Fever [54], Feverous [55], and Vitaminc [56]. For arithmetic reasoning, we test on the SVAMP dataset [57]. Our choice of tasks was driven by the performance of the models using the ToT method, which showed improvements in QA, fact verification, and arithmetic reasoning. We use 4-shot prompting for each dataset, with CoT demonstrations manually constructed by previous works [1, 17, 2]. Detailed experimental configurations can be found in Appendix A. For evaluation metrics, we report the accuracy and the average latency of generating the answer per instance. More metrics can be found in Appendix B.

**Baselines.** To validate the effectiveness of our proposed CPO, we consider the following baselines: 1) CoT [1], which prompts the LLM to generate a series of reasoning steps before producing the final answer. In our experiments, we use CoT with greedy decoding to assess the model's reasoning capabilities without any tuning. 2) ToT [8], which requires the LLM to explore multiple reasoning paths via tree search before generating the final answer. We use ToT to select reasoning paths and construct datasets to improve LLM's reasoning ability in the following TS-SFT baseline and our CPO method. 3) TS-SFT [11], which finds reasoning paths through tree search (i.e., ToT in our implementation) and then uses these paths during the supervised fine-tuning (SFT) process (referred to as SFT in Section 5.3 and 6).

**Implementation Details** Our experiments are based on widely used LLMs, specifically LLaMA2-7B/13B [14] and Mistral-7B [15]. For efficient fine-tuning, we use Low-Rank Adaptation (LoRA) adapters [58]. In all experiments, we set the regularization controller $\beta$ to $0.1$, generate $10$ new thoughts for each state, and retain the top $5$ thoughts after pruning at each step of reasoning. The temperature is set to $0.9$ for SVAMP and $0.4$ for the other datasets. The learning rates for DPO and SFT are 5e-6 and 1e-5, respectively. We use a batch size (with accumulation) of $32$ and optimize the LLM with AdamW [59]. For LoRA, the rank is set to $8$, and $\alpha$ is set to $16$. All experiments are conducted on NVIDIA A100 GPUs. The latency reported in Table 1 is based on a single NVIDIA A100 40GB. Both training and inference are performed using the Accelerate [60] backend. We train the LLMs for 4 epochs with early stopping based on the performance on a randomly sampled validation set. To mitigate the influence of randomness, all experiments are repeated three times with different random seeds, and the average results are reported.

### 5.2 Overall Results on Reasoning

Table 1 summarizes the performance across various reasoning tasks. We have the following findings:

**CPO improves LLM's reasoning ability.** As shown in Table 1, CPO enhances the reasoning ability of the base LLM, achieving an average improvement of $4.3\%$ and a maximum improvement of $9.7\%$ across all tasks and LLMs compared to the CoT approach. This indicates that CPO effectively improves the LLM's reasoning capabilities. Notably, CPO achieves these improvements without requiring additional human-annotated data, which is particularly beneficial in resource-constrained settings.

**CPO has a lower latency than ToT but comparable performance.** Although ToT consistently improves performance over CoT, it incurs high latency due to the need to generate and evaluate multiple thoughts at each reasoning step during inference. This process produces numerous tokens, resulting

Table 1: Experimental results for ToT, CoT, TS-SFT, and our proposed CPO across complex tasks including question answering, fact verification, and arithmetic reasoning are presented. $^*$ means significantly better than the best baseline (TS-SFT) with $p < 0.01$. **Bold** denotes the best method, and the second best if the top method is ToT.

| | | ToT [8] | | CoT [1] | | TS-SFT [11] | | CPO (ours) | |
|---|---|---|---|---|---|---|---|---|---|
| | | Acc. (%)↑ | Latency (s/ins.)↓ | Acc. (%)↑ | Latency (s/ins.)↓ | Acc. (%)↑ | Latency (s/ins.)↓ | Acc. (%)↑ | Latency (s/ins.)↓ |
| ***LLaMA2-7B*** | | | | | | | | | |
| *Question Answering* | Bam. | **33.6** | 1168.4 | 29.6 | 37.2 | 30.4 | 36.5 | **32.0**$^*$ | 38.2 |
| | 2Wiki. | 28.6 | 847.6 | 26.3 | 35.7 | 27.6 | 35.5 | **29.7**$^*$ | 35.7 |
| | Hot. | 23.0 | 1100.7 | 21.0 | 45.5 | 22.7 | 44.8 | **24.0**$^*$ | 41.1 |
| *Fact Verification* | FVR. | 47.3 | 2087.1 | 45.8 | 33.8 | 47.5 | 34.0 | **53.2**$^*$ | 36.8 |
| | FVRS. | 47.5 | 2539.5 | 44.3 | 40.6 | 46.0 | 40.4 | **49.0**$^*$ | 41.2 |
| | Vita. | 50.7 | 2639.3 | 47.3 | 35.9 | 51.0 | 40.1 | **52.7**$^*$ | 40.1 |
| *Arithmetic* | SVA. | 42.7 | 1861.1 | 37.7 | 33.3 | 43.1 | 30.2 | **46.0**$^*$ | 32.1 |
| *Average Performance* | | 39.1 | 1749.1 | 36.0 | 37.4 | 38.3 | 37.4 | **40.9**$^*$ | 37.9 |
| ***LLaMA2-13B*** | | | | | | | | | |
| *Question Answering* | Bam. | **53.8** | 1318.3 | 48.0 | 50.5 | 50.8 | 49.6 | **52.0**$^*$ | 50.3 |
| | 2Wiki. | **36.3** | 1097.1 | 28.3 | 67.0 | 29.0 | 66.5 | **30.3**$^*$ | 60.4 |
| | Hot. | **32.0** | 1271.0 | 29.0 | 65.2 | **30.3** | 65.5 | 30.3 | 63.8 |
| *Fact Verification* | FVR. | 48.8 | 3139.8 | 48.2 | 45.4 | 48.8 | 44.0 | **49.2** | 43.9 |
| | FVRS. | **51.3** | 3433.2 | 50.0 | 61.1 | 48.8 | 58.0 | **50.7**$^*$ | 68.2 |
| | Vita. | 52.5 | 2933.6 | 46.3 | 48.3 | 49.7 | 51.5 | **54.0**$^*$ | 58.6 |
| *Arithmetic* | SVA. | 45.7 | 2115.3 | 40.3 | 46.2 | 44.6 | 46.4 | **50.0**$^*$ | 48.1 |
| *Average Performance* | | **45.8** | 2186.9 | 41.4 | 54.8 | 43.1 | 54.5 | **45.2**$^*$ | 56.2 |
| ***Mistral-7B*** | | | | | | | | | |
| *Question Answering* | Bam. | **46.4** | 4399.6 | 41.6 | 46.2 | 41.6 | 45.0 | **45.6**$^*$ | 47.0 |
| | 2Wiki. | 28.4 | 2356.9 | 26.7 | 45.1 | 31.0 | 44.2 | **31.7** | 46.3 |
| | Hot. | **30.0** | 4698.0 | 28.0 | 58.4 | 28.6 | 56.2 | 29.4 | 56.9 |
| *Fact Verification* | FVR. | **61.4** | 3291.3 | 57.9 | 41.8 | **60.2** | 41.7 | 59.9 | 40.6 |
| | FVRS. | 50.5 | 5537.8 | 48.0 | 51.0 | 49.7 | 49.5 | **54.0**$^*$ | 53.0 |
| | Vita. | 52.2 | 4698.2 | 47.7 | 44.8 | 50.3 | 45.9 | **53.7**$^*$ | 46.6 |
| *Arithmetic* | SVA. | 66.0 | 4623.7 | 65.3 | 41.4 | 59.0 | 41.3 | **69.3**$^*$ | 44.9 |
| *Average Performance* | | 47.8 | 4229.4 | 45.0 | 47.0 | 45.8 | 46.3 | **49.1**$^*$ | 47.9 |

in significant computational and memory overhead [61]. In contrast, CPO shifts this computational burden to the training phase, maintaining the low latency of CoT (i.e., $57.5\times$ faster than ToT on average) during inference while providing comparable or superior performance. This demonstrates that our CPO can deliver enhanced reasoning capabilities without compromising efficiency.

**CPO surpasses TS-SFT on average.** Despite both CPO and TS-SFT using ToT to generate training data (where our implementation of ToT remains consistent), CPO exhibits an average improvement of 2.7% and reaches a maximum increase of 10.3%. A key factor behind this performance is the CPO's ability to fully utilize ToT's reasoning process. Specifically, CPO effectively leverages both selected and unselected thoughts at each reasoning step, whereas TS-SFT only uses information from the selected paths, offering CPO with a clear advantage. A detailed discussion of the effectiveness of CPO is presented in Section 5.3.

## 5.3 Component-wise Evaluations

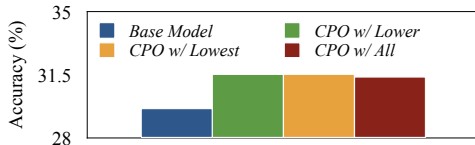
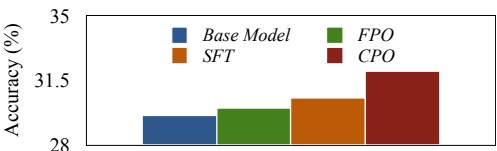

(a) Effect of dispreferred thoughts selection. *Base Model:* The base LLM. *CPO w/ Lowest:* CPO using the lowest-scoring thoughts as dispreferred. *CPO w/ Lower:* CPO using all lower-scoring thoughts as dispreferred. *CPO w/ All:* CPO using all thoughts not in the selected paths as dispreferred.

(b) Effect of per-step preference supervision. *Base Model:* The LLM without any fine-tuning. *SFT:* SFT of the base LLM (i.e., our TS-SFT baseline). *FPO:* DPO of the base LLM on paired full paths. *CPO:* CPO of the base LLM on paired per-step thoughts.

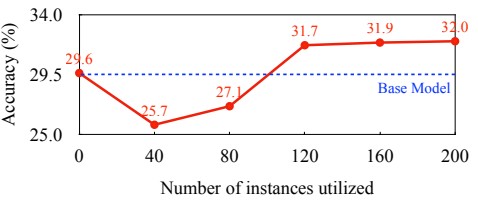
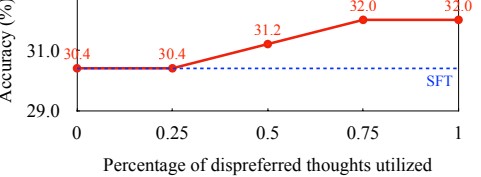

(c) Effect of the number of instances in generating paired thoughts. The number indicates the number of instances (e.g., questions in the QA task) used to generate paired thoughts for our CPO.

(d) Effect of dispreferred thoughts in optimization. The percentage indicates the proportion of data used for CPO, which uses dispreferred thoughts.

Figure 3: Component-wise evaluations and analysis on the Bamboogle dataset using the LLaMA2-7B as the base model.

**Effect of selection methods of dispreferred thoughts.** We analyze the impact of different methods for selecting dispreferred thoughts on model performance. As shown in Figure 4, we experiment with three strategies based on evaluation scores for each thought: 1) *CPO w/ Lowest:* Only the lowest-scoring thoughts in each reasoning step are dispreferred thoughts. 2) *CPO w/ Lower:* Thoughts with evaluation scores lower than the selected paths are dispreferred thoughts. 3) *CPO w/ All:* All thoughts not in the selected paths are considered dispreferred thoughts. We ensured an equal number of training samples for each strategy. Note that the evaluation score at each intermediate reasoning step (apart from the final one) determines whether to create the next reasoning step but not which thoughts are preferred. For example, as shown in the figure, even though the score of 32 is higher than 23, the thought with a score of 23 is preferred since it is part of the selected path.

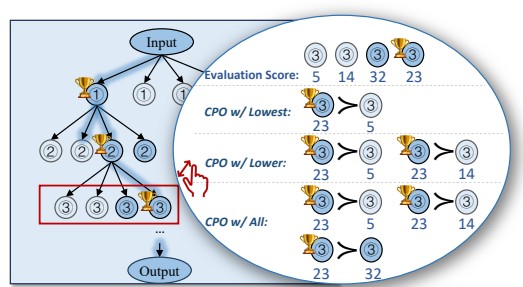

Figure 4: Different strategies for selecting dispreferred thoughts and their impact on model performance. At each reasoning step, three strategies are used to select dispreferred thoughts based on their reasoning scores: 1) *CPO w/ Lowest:* Selects only the thought with the lowest score. 2) *CPO w/ Lower:* Selects all thoughts with scores lower than the preferred thought. 3) *CPO w/ All:* Selects all thoughts as dispreferred as long as they are not the preferred thought.

The results in Figure 3(a) show that the performance differences among these strategies are minimal. This suggests that the distinction between preferred and dispreferred thoughts is better determined in the selected reasoning path rather than intermediate evaluation scores. To obtain a greater number of preferred thoughts for each instance to create paired preference thoughts, we chose the *CPO w/ All* strategy.

**Effect of numbers of training data.** To assess the impact of the number of training data used in optimization, we conduct an ablation analysis by varying the number of instances (e.g., questions in the QA task) used to generate paired preference thoughts, ranging from 0 to 200. As illustrated in

Table 2: Effect of different kinds of training data on the Bamboogle dataset using the LLaMA2-7B as the base model.

| Data | Description | SFT | CPO |
|------|-------------|-----|-----|
| Single-Task | Training only on specific task (Bamboogle) data. | 30.4 | **32.0** |
| Uniform QA | Training on 3 datasets of the same type (QA) as the test task. | 31.2 | **35.2** |
| Mixed-Type | Training on all 7 different types of data. | 29.6 | **35.2** |

Figure 3(c), we observe that with an increase in the number of instances, the model's performance initially declines and then rises. Specifically, when trained with data generated from less than 80 instances, the model's performance is even worse than without any training, likely due to overfitting [62], which leads to performance degradation. However, as the number increases to 120, the model's performance consistently improves. Optimizing with paired thoughts from 120 instances, the model's performance surpasses that of the base model. When the number exceeds 120, the model's performance converges, indicating a balance of data for training.

**Sensitivity of data mixture.** We explore the performance of the CPO method across diverse data settings to assess its adaptability and learning efficiency from various data types. As shown in Table 2, we specifically examine three different data configurations: 1) single task data, 2) uniform QA data, and 3) mixed-type data. Our findings indicate that CPO demonstrates performance improvements of 3.2% in both settings 2 and 3, suggesting its robust ability to harness diverse data sources to enhance learning outcomes. In contrast, the SFT method exhibits comparable performance across these settings, indicating a different sensitivity to data diversity. It is worth noting that, to ensure fairness, although we find that mixed data leads to better performance, the experiments in Table 1 are conducted using individual datasets for training, consistent with the baselines.

# 6 Analysis

**Do we need dispreferred information?** We explore the impact of dispreferred thoughts on model performance by gradually incorporating these thoughts into the training data. Initially, we introduce dispreferred thoughts for their corresponding preferred counterparts and apply CPO to this segment of the data. For preferred thoughts without dispreferred counterparts, we implement SFT on these data. Consequently, the percentage of dispreferred thoughts incorporated can also be viewed as the proportion of data processed using CPO. We adjust the inclusion percentage of dispreferred thoughts from 0% to 100%. An inclusion of 0% indicates that we utilize SFT solely on the preferred thoughts, i.e., the baseline TS-SFT. Conversely, an inclusion of 100% signifies our CPO, where the entire dataset includes paired preferred and dispreferred thoughts.

**Why is chain level optimization important?**
As shown in Figure 3(d), we find that increasing the percentage of dispreferred data inclusion consistently improves model performance. This suggests that dispreferred thoughts are beneficial during the optimization process, highlighting the importance of leveraging both preferred and dispreferred thoughts for enhancing the model's reasoning capabilities.

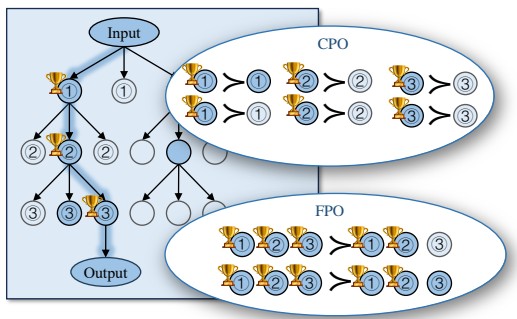

Unlike our CPO, an alternative approach is to construct preference data using complete reasoning paths, i.e., using the selected full reasoning paths as preferred and other paths as dispreferred data, as shown in Figure 5. This method essentially applies DPO at the full-path level, referred to here as Full-path Preference Optimization (FPO). However, FPO encounters a significant issue where the gradients of the longest common prefix (LCP) tokens in paired data cancel

Figure 5: Illustrations of two different ways to construct paired preference data: 1) CPO: Paired preference data are constructed at each thought step. 2) FPO: Paired preference data are constructed only at the full path level.

out, which we call the *LCP gradient cancellation* issue. For example, for the preferred path $\hat{y}_w = [5, +, 4, =, 9, and, 9, +, 2, =, 11]$ and the dispreferred path $\hat{y}_l = [5, +, 4, =, 9, and, 9, +, 2, =, 15]$, the gradient will only be computed for the last token where the two sequences diverge.

To mathematically illustrate how LCP gradient cancellation happens in FPO, consider $\hat{y}_w = [p_{1:n}, w_{n+1}]$ and $\hat{y}_l = [p_{1:n}, l_{n+1}]$, where $p$ is the longest common prefix sequence between $\hat{y}_w$ and $\hat{y}_l$. The gradient of FPO is given by:

$$\nabla_\theta \mathcal{L}_{\text{FPO}}(\pi_\theta; \pi_{\text{ref}}) = C(\theta) \cdot \nabla_\theta \left( \log \pi_\theta(\hat{y}_w|x) - \log \pi_\theta(\hat{y}_l|x) \right)$$

$$= C(\theta) \cdot \nabla_\theta \left( \boxed{\log \pi_\theta(p_{1:n}|x)} + \log \pi_\theta(w_{n+1}|x, p_{1:n}) - \boxed{\log \pi_\theta(p_{1:n}|x)} - \log \pi_\theta(l_{n+1}|x, p_{1:n}) \right),$$

where $C(\theta)$ is a scalar function that does not affect the direction of the gradient and can be absorbed into the learning rate.

We can clearly see that the gradient terms of the common prefix tokens (highlighted with boxes) cancel each other out. This issue also exists in DPO training [63], but FPO suffers more frequently and severely due to the longer LCP between paired data constructed by tree search. As an empirical evidence, we observe the LCP length accounts for $28\%$ of the total length in the Bamboogle dataset. CPO, on the other hand, constructs preference data at every step in the reasoning chain, allowing optimization of the LLM on all steps in the reasoning path. This means the common prefix can be optimized at its own step, ensuring that the gradient still exists for the common prefix.

We also compare FPO to CPO empirically in Figure 3(b), which further substantiates this observation. Switching to FPO led to a relative performance decrease of $4.6\%$, even worse than the baseline SFT that does not utilize any information from dispreferred data. This underscores the importance of per-step preference thoughts for CPO.

## 7 Conclusion

In this work, we introduce a novel method called Chain of Preference Optimization (CPO), which leverages the supervision generated by the self-reasoning process (i.e., tree-of-thoughts) to enhance the reasoning ability of LLMs. Experiments on three different LLMs across seven different datasets demonstrate that CPO can consistently improve the performance of the base model by $4.3\%$ on average without sacrificing inference speed. Furthermore, our method also substantially outperforms the strong baseline TS-SFT and even achieves comparable performance to the ToT method, which requires approximately $57.5$ times more inference time.

In future work, we plan to integrate CPO with other reasoning algorithms, such as Graph-of-Thoughts [9] and AlphaZero-like tree search [11]. Furthermore, we intend to explore the potential of using a weaker LLM to evaluate a stronger one within the CPO framework, facilitating weak-to-strong alignment [64].

## Acknowledgement

We thank Cunxiao Du for the insightful derivation of the Longest Common Prefix (LCP) gradient cancellation and for the valuable contributions to the discussions throughout this work. We also appreciate Fangkai Jiao's helpful feedback. Additionally, we thank the anonymous reviewers for their constructive comments and suggestions that helped improve this paper.

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

## Societal Impacts and Limitations

Since our CPO does not require any human annotation, it can be directly used. For example, to protect the safety of large models, one can simply provide a constitution, and then fine-tune the LLM to make it more compliant. This also introduces another issue: our method can be adjusted for malicious applications. Our limitation is that we still need to generate data through ToT, which is a time-consuming process. We aim to accelerate the complex reasoning processes during training data generation by incorporating methods like non-autoregressive generation [65, 66, 67], speculative decoding [68, 69], and KV cache pruning [70, 71, 72]. Additionally, we have only tested this on text language models and have not tried it on vision-language models. Furthermore, the application scope of our method remains restricted to a small set of downstream tasks. Expanding its application to diverse tasks, such as text classification [73], news verification [74, 75], machine translation [76], and software engineering [77], remains an area for future research. Moreover, ethical considerations must be taken into account, as the potential for misuse could lead to unintended consequences.

## A   Detailed Experiment Configurations

To maintain a reasonable budget, especially given the high computational demand of ToT, we limit each dataset to a maximum of 300 test samples through random sampling. For datasets that contain less than 300 test samples, we instead use all available samples. For training, we randomly select less than 300 instances from each dataset to construct the preference data pairs, **without** using the ground-truth labels. This is because we observe that more number of training data does not lead to performance improvement as shown in Section 5.3. In addition, in our experiments, approximately 200 samples (e.g., questions in the QA task) on average could generate about 6,531 preference pairs, suggesting that our CPO requires only a small number of samples by design. Constructing preference data is a time-intensive process. The choice of 300 samples represents a practical trade-off between efficiency and effectiveness, allowing us to manage resources effectively while still achieving noticeable improvements.

## B   Additional Experiments

**CPO benefits from iterative learning.**    Inspired by the iterative improvements achieved in previous research [28, 78], in this section, we explore whether CPO can be further improved by iterative learning. Specifically, we try two distinct iterative training strategies: *1. SFT+CPO*: in *iter=0*, Start with a base LLM that has not been fine-tuned at all; in *iter=1*, SFT the base LLM on the reasoning path selected by ToT (base model); in *subsequent iterations (iteration >1)*, Continue to fine-tune the model using the CPO method, based on the chain of preference thoughts constructed by the model in the previous iterations. and *2. CPO only*: in *iter=0*, same as *iter=0* in *SFT+CPO*; in *subsequent iterations (iteration >0)*: Only use the CPO method for training in all iterations, similar to the approach in *SFT+CPO* after the first iteration.

As shown in Table 3, We find that if use CoT for inference, as the number of iterations increases, the performance of the model gradually improves. In the *CPO only* setting, the performance improves by 4% after two iterations. However, an intriguing phenomenon is noted: if we use the ToT method for inference on our fine-tuned models, the performance does not consistently rise and sometimes even declines. For instance, in the *SFT+CPO* setting, after the first round of SFT, the performance with ToT decreased by 2.7%. We hypothesize this may be related to a decrease in the diversity of the model's outputs after fine-tuning, which reduces the search space for ToT, making it challenging to find better reasoning paths. When the performance of CoT and ToT becomes similar, further fine-tuning of the LLM leads to convergence in the *SFT+CPO* setting and even a decline in the *CPO only* setting.

**Comparison with ReST and self-rewarding baseline.**    The settings of ReST [33] and self-rewarding [28] are different from ours, as discussed in Section 2. These methods rely on either external reward models (ReST) or labeled data (self-rewarding), which makes them not directly comparable to our approach. To ensure a fair comparison with our CPO method, we prompted the LLM itself in the same way as our CPO to serve as the reward model for ReST and as the labeled data

| Inference Method | SFT+CPO | | | | CPO only | | | |
|---|---|---|---|---|---|---|---|---|
| | *iter=0* (*Base*) | *iter=1* (*SFT*) | *iter=2* (*CPO*) | *iter=3* (*CPO*) | *iter=0* (*Base*) | *iter=1* (*CPO*) | *iter=2* (*CPO*) | *iter=3* (*CPO*) |
| CoT | 29.6 | 30.4 | 30.4 | 31.2 | 29.6 | 32.0 | 33.6 | 31.2 |
| ToT | 33.6 | 30.4 | 31.2 | - | 33.6 | 34.4 | 33.6 | - |

Table 3: Effects of iterative learning on the Bamboogle dataset using the LLaMA2-7B as the base model.

annotator for self-rewarding, respectively. As shown in Table 4, the results indicate that, on average, our CPO method surpasses both ReST and self-rewarding under this fair comparison setting.

| Model | Bam. | 2wiki | Hotpotqa | Fever | Feverous | Vitaminc | Svamp | Average |
|---|---|---|---|---|---|---|---|---|
| ***LLaMA2-7b*** | | | | | | | | |
| *Rest* | 30.4 | 24.0 | 22.3 | 45.5 | 43.9 | 51.7 | 42.3 | 37.2 |
| *SelfR* | 31.2 | 25.3 | 21.0 | 48.8 | 44.7 | 51.3 | 43.0 | 37.5 |
| *CPO* | 32.0 | 29.7 | 24.0 | 53.2 | 49.0 | 52.7 | 46.0 | 40.9 |
| ***LLaMA2-13b*** | | | | | | | | |
| *Rest* | 48.0 | 28.3 | 28.7 | 46.8 | 48.0 | 50.2 | 44.3 | 42.0 |
| *SelfR* | 48.0 | 29.0 | 30.0 | 47.8 | 48.5 | 51.0 | 45.3 | 42.8 |
| *CPO* | 52.0 | 30.3 | 30.3 | 49.2 | 50.7 | 54.0 | 50.0 | 45.2 |
| ***Mistral-7b*** | | | | | | | | |
| *Rest* | 43.2 | 26.7 | 27.4 | 59.5 | 48.3 | 49.7 | 63.3 | 45.4 |
| *SelfR* | 44.0 | 28.0 | 28.1 | 58.2 | 48.0 | 50.0 | 65.0 | 45.9 |
| *CPO* | 45.6 | 31.7 | 29.4 | 59.9 | 54.0 | 53.7 | 69.3 | 49.1 |

Table 4: Performance comparison across different datasets.

**More metrics on QA dataset.** We include F1 scores for the three QA tasks in Table 5. The results show that the F1 performance aligns well with the corresponding accuracy for each task.

**Ablation studies across datasets and models.** Figure 6, 7, and 8 provide ablations and analysis across various models and datasets. The observed trends remain generally consistent across these different settings.

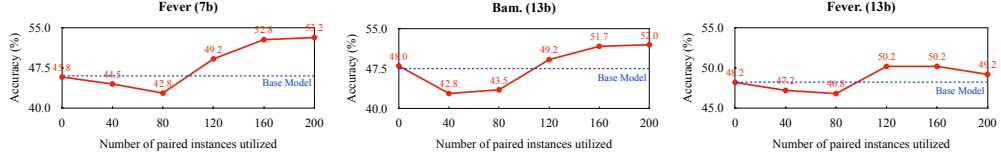

Figure 6: Effect of the number of instances in generating paired thoughts.

**Illustrative examples of the reasoning paths preferred by CPO.** Table 6 presents examples demonstrating that the paths favored by CPO align more closely with those selected by ToT than by CoT, indicating a higher reasoning quality in CPO's chosen paths.

|         | LLaMA2-7b |      |      |      | LLaMA2-13b |      |      |      | Mistral-7b |      |      |      |
|---------|-----------|------|------|------|------------|------|------|------|------------|------|------|------|
|         | ToT | CoT | SFT | CPO | ToT | CoT | SFT | CPO | ToT | CoT | SFT | CPO |
| Bam.    | 29.5 | 28.1 | 29.3 | 31.8 | 43.0 | 40.9 | 41.4 | 43.0 | 48.1 | 42.2 | 41.9 | 47.5 |
| 2wiki.  | 26.5 | 23.6 | 25.0 | 26.0 | 30.7 | 26.8 | 27.2 | 28.3 | 25.5 | 24.3 | 27.3 | 27.0 |
| hot.    | 23.4 | 20.5 | 22.9 | 24.7 | 33.3 | 31.3 | 31.3 | 32.0 | 29.9 | 25.9 | 26.8 | 28.0 |

Table 5: F1 scores on QA datasets.

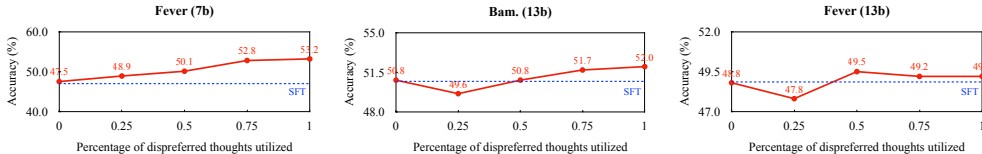

Figure 7: Effect of dispreferred thoughts in optimization.

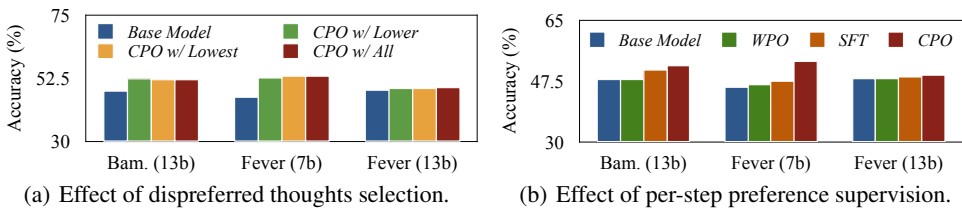

(a) Effect of dispreferred thoughts selection.  (b) Effect of per-step preference supervision.

Figure 8: Component-wise Evaluations and Analysis.

| **Question 1**: When did the author of Annabel Lee enlist in the army? |
|---|
| **CoT**: Step 1, when did Edgar Allan Poe enlist in the army? Edgar Allan Poe enlisted in the army on May 27, 1827. Step 2, when did Edgar Allan Poe die? Edgar Allan Poe died on October 7, 1849. Step 3, so the final answer is: October 7, 1849. |
| **ToT**: step 1, when did edgar allan poe enlist in the army? edgar allan poe enlisted in the army on august 16, 1827. step 2, what year did edgar allan poe enlist in the army? edgar allan poe enlisted in the army in 1827. step 3, so the final answer is: 1827. |
| **CPO**: step 1, when did Edgar Allan Poe die? Edgar Allan Poe died on October 7, 1849. Step 2, when did Edgar Allan Poe enlist in the army? Edgar Allan Poe enlisted in the army on May 27, 1827. Step 3, so the final answer is: May 27, 1827. |

| **Question 2**: What is the nickname of the easternmost U.S. state? |
|---|
| **CoT**: Step 1, who is the scientist at MIT that won the Queen Elizabeth Prize for Engineering in 2013? The scientist at MIT that won the Queen Elizabeth Prize for Engineering in 2013 was Daniela Rus. Step 2, who is the father of Daniela Rus? Daniela Rus's father is Vladimir Rus. Step 3, so the final answer is: Vladimir Rus. |
| **ToT**: step 1, what is the name of the easternmost u.s. state? maine is the easternmost u.s. state. step 2, what is the nickname of maine? maine's nickname is 'the pine tree state. |
| **CPO**: Step 1, what is the name of the easternmost U.S. state? The easternmost U.S. state is Maine. Step 2, what is the nickname of Maine? The nickname of Maine is The Pine Tree State. Step 3, so the final answer is: The Pine Tree State. |

Table 6: Illustrative examples of the reasoning paths preferred by CPO. Similar paths between ToT and CPO are in yellow.

