# OpenReview forum: "Chain of Preference Optimization: Improving Chain-of-Thought Reasoning in LLMs"
_NeurIPS.cc/2024/Conference — NeurIPS 2024 poster_

### Official Review · Reviewer_iCyz · 2024-06-30

**Soundness:** 3
**Presentation:** 3
**Contribution:** 3
**Rating:** 6
**Confidence:** 4

**Summary:**

The paper introduces Chain-of-Preference Optimization (CPO) to improve reasoning in LLMs. CPO utilizes preference data generated during ToT to fine-tune models. This approach improves reasoning without increasing inference complexity.

**Strengths:**

1. The idea of CPO is interesting.
2. CPO achieves enhanced performance without increasing inference complexity.
3. The paper is well-written and easy to follow.

**Weaknesses:**

1. While CPO is effective across several tasks, its performance on more complex reasoning benchmarks, such as the MATH benchmark, remains unexplored.
2. Fine-tuning with reasoning paths generated by ToT may reduce the diversity of generated paths. A systematic investigation into how the generated reasoning paths change before and after fine-tuning is needed.
3. The effectiveness of CPO is heavily dependent on the quality of reasoning paths generated by ToT. If ToT produces suboptimal paths, the preference data might not be as beneficial.
4. The paper does not adequately address the applicability of CPO to more advanced models, such as LLaMA3.
5. While CPO enhances performance, the paper lacks a deep analysis of the interpretability of the optimized reasoning paths. Understanding why certain paths are preferred could provide additional insights and improve trust in the model’s decisions.

**Questions:**

NA

**Limitations:**

Yes

---

> ### Author Rebuttal · Authors · 2024-08-07
>
> Thank you for your valuable feedback and questions. Below, we respond to the comments in ***Weaknesses (W)*** and ***Questions (Q)***.
>
> ---
>
> ***W1: CPO performance on MATH benchmark.***
>
> Following your feedback, we included the performance of both CoT and CPO using the LLaMa3-8b-base model in $\\textrm{\\color{blue}Table B}$ of the Rebuttal PDF. For the MATH benchmark, we tested on a sample of 300 instances, as discussed in our response to  ***W1*** of **Reviewer VaVd**. As indicated in the table, CPO enhances performance by 1.1% on the MATH benchmark relative to CoT, demonstrating its effectiveness on more complex reasoning tasks.
>
> ---
>
> ***W2: CPO's influnce on the diversity of generated paths.***
>
> In response to your insightful suggestion, we systematically evaluated the diversity of generated reasoning paths before and after fine-tuning, as measured by distinct-N [H]. The results are summarized in the table below:
>
> ||LLaMA2-7b||||	LLaMA2-13b||||	Mistral-7b||||
> |  ----  | ----  | ----  | ----  | ----| ----  | ----| ---- | ----| ----  | ----| ----  | ----|
>  ||distinc-1|distinc-2|distinc-3|distinc-4|distinc-1|distinc-2|distinc-3|distinc-4|distinc-1|distinc-2|distinc-3|distinc-4|
> |Before|0.12|0.408|0.568|0.632|0.035|0.167|0.288|0.366|0.460|0.199|0.329|0.407
> |After|0.039|0.148|0.229|0.287|0.032|0.144|0.272|0.35|0.037|0.134|0.215|0.271|
>
> The results indicate a decrease in distinct-N values after fine-tuning with CPO, suggesting a reduction in diversity. This phenomenon may stem from our preference optimization algorithm (i.e., DPO), which optimizes the reverse KL divergence and inherently exhibits mode-seeking behavior, thus reducing diversity in generation [I]. We will incorparate this evaluation and analysis in the revision.
>
> [H] Li et al. A diversity-promoting objective function for neural conversation models, NAACL 2016
>
> [I] Wang et al. Beyond Reverse KL: Generalizing Direct Preference Optimization with Diverse Divergence Constraints, arXiv:2309.16240
>
>
> ---
>
> ***W3: CPO's effectiveness is heavily dependent on ToT's performance.***
>
> We acknowledge the concerns raised regarding the dependence of CPO on the quality of reasoning paths generated by ToT. While it is true that suboptimal paths from ToT could affect the overall performance, our results indicate that CPO has the capability to occasionally surpass the performance upper bounds of ToT. This is evidenced by the results presented in Table 1 of our paper.
>
> Additionally, CPO's design is not inherently restricted to using ToT-generated paths. Our framework is adaptable and can integrate with other tree search methods that might yield higher quality reasoning paths, which we leave as our future work.
>
> ---
>
> ***W4: CPO's performance on LLaMA3.***
>
> The release of LLaMA3 on April 18, 2024, just 28 days prior to our submission deadline, presented significant challenges in conducting comprehensive experiments across all seven tasks with this new model.
>
> In response to your feedback, we have included additional results for the LLaMA3-8b-base model’s performance using both CoT and CPO on the ProofWriter and MATH benchmarks in Table B of the Rebuttal PDF. These results demonstrate that CPO enhances performance by 1.1% on both datasets when compared to CoT, affirming its applicability even on more advanced models.
>
> ---
>
> ***W5: The interpretability of the optimized reasoning paths.***
>
> We appreciate your emphasis on the importance of interpretability in the reasoning paths optimized by CPO. To address this, we have included an analysis using two specific examples in  $\\textrm{\\color{blue}Table E}$ of the Rebuttal PDF. These examples illustrate that the paths preferred by CPO closely resemble those selected by ToT rather than CoT, suggesting a relatively higher quality of reasoning in the paths chosen by CPO. A thorough analysis of these patterns and their implications will be provided in the revision to enhance the interpretability and trustworthiness of the model's decision-making process.

---

### Official Review · Reviewer_ADqw · 2024-07-05

**Soundness:** 3
**Presentation:** 3
**Contribution:** 2
**Rating:** 5
**Confidence:** 4

**Summary:**

This paper introduces Chain of Preference Optimization (CPO), a method to enhance mathematical reasoning in large language models by feeding step-level pairs rather than response-level ones into DPO objectives. CPO leverages non-optimal reasoning thoughts from tree-search processes to construct paired preference data at each reasoning step, where both responses share the same content until a certain step and then start varying. Experiments across seven datasets show CPO significantly improves performance compared to Chain-of-Thought approaches, achieving comparable results to Tree-of-Thought methods while being substantially faster during inference. The authors provide a comprehensive analysis of CPO's components and demonstrate its effectiveness in utilizing both preferred and dispreferred thoughts in the reasoning process.

**Strengths:**

- The experiments demonstrate that CPO is effective across different tasks including question answering, fact verification, and arithmetic.
- This paper conducts an extensive analysis of several factors that can influence the model performance. The insights on scaling effects, data mixture, and dispreferred information can be useful.
- The paper is well-written and easy to follow.

**Weaknesses:**

- The major concern is that despite being effective, the method is basically a new use of the DPO algorithm with different inputs (converting response-level pairs to chain-level) and thus it's fundamentally still DPO.
- In section 6, the authors have explained why chain-level optimization is important. However, according to the explanation, token-level DPO is more natural in locating errors and avoiding LCP gradient cancellation. Then according to your theory, why not directly use token-level optimization, which is exactly what PPO does, but an intermediate chain-level? I suppose there exists an efficiency-effectiveness trade-off in these settings.

**Questions:**

See weakness.

**Limitations:**

See above

---

> ### Author Rebuttal · Authors · 2024-08-07
>
> Thank you for your valuable feedback and questions. Below, we respond to the comments in ***Weaknesses (W)*** and ***Questions (Q)***.
>
> ---
>
> ***W1: CPO is fundamentally still DPO.***
>
>
> Thank you for raising this point. We would like to clarify that while our approach adopts the DPO algorithm to fine-tune language models, our core contribution lies in different aspects compared to DPO.
>
> DPO is an algorithm designed to align models (typically LLMs) with a target reward directly through pair preference data, avoiding the need for a surrogate reward model, and is widely used in RLHF for LLMs. The algorithm itself does not specify how to obtain the preference data, which can be sourced from human annotators [26,41,5,40], external reward models [43,1], or the model being fine-tuned itself [14].
>
> Our work falls into a specific methodology for constructing preference data from the model itself. We find that time-consuming inference schemes such as ToT naturally produce thought-level preference data that are helpful for learning more time-efficient inference methods like CoT. This construction, although cost-effective (as it requires no external reward models or human annotation), has not been explored in previous research. Please also refer to our response to ***W1*** of **Reviewer WuBE** for a discussion and comparison with closely related works.
>
> Moreover, the DPO algorithm in our approach is not the only choice. Advanced preference optimization algorithms (such as SimPO [F]) could potentially yield better results, which we will leave as future work. Please also refer to our response to ***Q1*** of **Reviewer WuBE**.
>
> [F] Meng et al. SimPO: Simple Preference Optimization with a Reference-Free Reward, arXiv:2405.14734.
>
> ---
>
> ***W2: Why not directly use token-level optimization.***
>
> We opted not to use token-level optimization primarily due to the following reasons:
>
> 1. Challenges in Reward Acquisition: CPO relies on constructing preference data from the model’s self-evaluation capabilities. Our preliminary experiments suggested significant difficulties in generating reliable token-level rewards using prompt-based LLM methods, which is crucial for token-level preference data construction.
> 2. High Computational Cost: The computational demands and associated costs of token-level ToT are substantially greater than those for thought-level ToT.
>
> Considering these factors, chain-level optimization presents a balanced approach, mitigating both the computational overhead and the LCP gradient cancellation problem.

---

### Official Review · Reviewer_WuBE · 2024-07-11

**Soundness:** 3
**Presentation:** 3
**Contribution:** 3
**Rating:** 6
**Confidence:** 4

**Summary:**

This paper presents a method called Chain of Preference Optimization (CPO) that fine-tunes large language models (LLMs) using the tree-of-thought (ToT) method to improve the performance of chain-of-thought (CoT) decoding. CPO aligns each step of CoT reasoning paths with those of ToT by leveraging preference information in the tree-search process, resulting in similar or better performance than CoT alone while avoiding the significant inference burden. Experimental results demonstrate its effectiveness.

**Strengths:**

Innovation: The method of constructing preference dataset with ToT has a certain novelty.

Writing quality:  Your paper is well-written, with clear and precise language and a smooth flow of ideas.  The structure is reasonable,  and the logic is sound, making it very enjoyable to read.

Experimental analysis: This paper has made a full experimental demonstration.

**Weaknesses:**

1.  The experiment lacked a stronger baseline comparison.  In my opinion, the author's method belongs to the self-imporvement methods, so it should be compared with such typical methods, such as the ReST and self-rewarding mentioned by the author in related work.

2.  The ToT approach to dataset construction appears to be computationally expensive.

3.  the author does not seem to provide a task-specific prompt templates (although the general guideline prompt is mentioned in line 141).  This is not necessary, but I think the quality of building a preference dataset can be seriously affected by the prompt template of evaluation.

**Questions:**

1. Why use DPO instead of other preference optimization algorithms?
2. Explain why the experiment lacks different self-imporvement baseline algorithms.
3. There may be a large number of negative samples in the preference dataset constructed by ToT making the dataset unbalanced.  May I ask whether this will have a negative impact on the training stage?

**Limitations:**

The limitations of the study and the possible negative social impact have been well documented by the authors.

---

> ### Author Rebuttal · Authors · 2024-08-07
>
> Thank you for your valuable feedback and questions. Below, we respond to the comments in ***Weaknesses (W)*** and ***Questions (Q)***.
>
> ---
>
> ***W1: Comparision with ReST and self-rewarding baseline***
>
> We would like to clarify that the settings of ReST and self-rewarding are different from ours, as discussed in L66-L78 of Section 2. These methods rely on either external reward models (ReST) or labeled data (self-rewarding), which makes them not directly comparable to our approach.
>
> However, following your suggestion, we have added a comparison with these two baselines in the table below. To ensure a fair comparison with our CPO method, we prompted the LLM itself in the same way as our CPO to serve as the reward model for ReST and as the labeled data annotator for self-rewarding, respectively. The results indicate that, on average, our CPO method surpasses both ReST and self-rewarding under this fair comparison setting.
>
> We will include the two baselines in the revision.
>
>
> |||Bam.|	2wiki.|	Hotpotqa|	Fever|	Feverous|	Vitaminc|	Svamp|Average|
> |  ----  | ----  | ----  | ----  | ----  | ----  | ----  | ----  | ----  |----  |
> |LLaMA2-7b|	Rest|	30.4|24.0|22.3|45.5|43.9|51.7|42.3|37.2|
> ||	SelfR|	31.2|	25.3|	21.0|	48.8|	44.7|	51.3|43.0|37.5|
> ||	CPO|	**32.0**|	**29.7**|	**24.0**|	**53.2**|	**49.0**|	**52.7**|**46.0**|**40.9**|
> |LLaMA2-13b|	Rest|48.0|28.3|28.7|46.8|48.0|50.2|44.3|42.0|
> ||	SelfR|48.0|29.0|30.0|47.8|48.5|51.0|45.3|42.8|
> ||	CPO|**52.0**|**30.3**|**30.3**|**49.2**|**50.7**|**54.0**|**50.0**|**45.2**|
> |Mistral-7b|	Rest|43.2|26.7|27.4|59.5|48.3|49.7|63.3|45.4|
> ||	SelfR|44.0|28.0|28.1|58.2|48.0|50.0|65.0|45.9|
> ||	CPO|**45.6**|**31.7**|**29.4**|**59.9**|**54.0**|**53.7**|**69.3**|**49.1**|
>
>
> ---
>
> ***W2: Dataset construction using ToT is computationally expensive.***
>
> The computational expense introduced by ToT is limited to the training stage. A key motivation of our approach is to transfer the computational burden from the inference to the training phase. This allows CPO to directly produce answers through greedy decoding during inference, significantly enhancing efficiency.
>
> Moreover, by using only 200 samples to generate preference pairs, we were able to achieve performance improvements efficiently (Please also refer to our response to ***Q3*** of **Reviewer VaVd**).
>
> ---
> ***W3: Task-specific evaluation prompt templates***
>
> We apologize for any confusion. To clarify, the prompt template used in our evaluation consists of two parts: (1) the general guidelines, and (2) task-specific demonstration examples (L135-L137).
>
> For example, here is one of the demonstration examples we used for QA datasets:
>
> Question: When did the last king from Britain's House of Hanover die?
> Thought: Step 1, when did the last king from Britain's House of Hanover born?
> Evaluation Process:
> The thought process focuses on the birth date of the last king from Britain's House of Hanover. However, knowing the birth date does not directly help in determining the date of death, which is the actual question. The lifespan of an individual can vary widely and cannot be accurately inferred from their birth date alone. Therefore, this thought process is unlikely to lead to the correct answer without additional information.
> So the evaluation result is: this thought is impossible to help pariticially or directly answer the question.
> Evaluation Results:
> Impossible
>
> We will clarify theis and provide task-specific demonstration examples in the revision.
>
>
> ---
>
> ***Q1: Choice of DPO instead of other preference optimization algorithms.***
>
> Our contribution primarily focuses on the construction of preference data (Please also refer to our response to ***W1*** of **Reviewer ADqw**). We chose to experiment with DPO because it is one of the most typical and widely used preference optimization algorithms. While we acknowledge that DPO is not the sole option available, advanced algorithms like SimPO [F] could potentially yield superior results, which we will leave as future work.
>
> [F] Meng et al. SimPO: Simple Preference Optimization with a Reference-Free Reward, arXiv:2405.14734.
>
> ---
>
> ***Q2: Lacking self-imporvement baselines.***
>
> Please see our response to ***W1***.
>
> ---
>
> ***Q3: Unbalanced preference dataset constructed by ToT.***
>
> In the constructed preference data, each positive sample is structurally paired with a corresponding negative sample, maintaining a one-to-one mapping at the level of individual pairs, thus achieving balance structurally. However, since ToT selects more paths than it rejects, the overall dataset exhibits an imbalanced distribution with more positive than negative samples.
> Despite this numerical discrepancy, our method oversamples positive instances by design, which could mitigate the potential negative impacts of an unbalanced dataset on the CPO model training. This strategy is also adopted by [G]. Figure 3(a) of our paper further substantiates this hypothesis by exploring various methods for selecting negative samples, which leads to differerent quantities of negative examples. The results in the figure suggests that the number of negative samples has minimal impact on the training outcomes.
>
> [G] Pattnaik et al. Curry-DPO: Enhancing Alignment using Curriculum Learning & Ranked Preferences, arXiv:2403.07230.

---

> > ### Comment · Reviewer_WuBE · 2024-08-09
> >
> > Thanks to the author for solving my confusion. The experiments added by the author have well solved my worries, so I raised my score from 4 to 6.

---

> > > ### Author Response · Authors · 2024-08-09
> > >
> > > We greatly appreciate your valuable feedback and the score improvement. We will further polish the paper and incorporate the rebuttal discussions into the final revision. Thank you!

---

### Official Review · Reviewer_Nj7W · 2024-07-12

**Soundness:** 3
**Presentation:** 3
**Contribution:** 3
**Rating:** 6
**Confidence:** 4

**Summary:**

The paper presents a novel method to enhance the performance of LLMs in complex problem-solving tasks using Chain of Preference Optimization (CPO). The authors propose fine-tuning LLMs using the search tree constructed by tree-of-thought (ToT), allowing CoT to achieve similar or better results without the heavy tree-searching during inference. The CPO method aligns each step of the CoT reasoning paths with those of ToT, leveraging the inherent preference information in the tree-search process. The authors provide experimental evidence of the effectiveness of CPO, showing significant improvements in LLM performance in tasks such as question answering, fact verification, and arithmetic reasoning.

**Strengths:**

The paper presents a novel method, Chain of Preference Optimization (CPO), that significantly enhances the CoT reasoning ability of LLMs without increasing the inference load. It offers a convincing comparison with the ToT method, demonstrating that CPO substantially reduces inference time, as evidenced by the decrease in latency from 1749s/ins to 38s/ins on Llama2-7B. Furthermore, the paper shows that CPO not only matches but surpasses the ToT baseline in terms of accuracy across multiple datasets. This indicates that CPO effectively fine-tunes LLMs to generate more optimal reasoning paths, thereby improving their performance in complex problem-solving tasks. The research is significant as it provides a practical solution to the challenge of balancing inference complexity and reasoning quality in LLMs.

**Weaknesses:**

The paper presents a promising approach to fine-tuning LLMs using CPO. However, there is a significant gap in the evaluation of the potential impacts of this fine-tuning process on other abilities of the pretrained LLMs. This omission could leave readers questioning the broader implications and versatility of the proposed method. Future work should consider a comprehensive evaluation of the fine-tuning process to fully understand its effects on the LLMs' overall performance.

**Questions:**

LN35, what is the advantage of CPO comared to existing MCTS methods? And why?

LN117, have you considered online RL that regenerates the chain-of-preferences using the updated LLM?

Table1, I'm wondering how does the method perform on common reasoning tasks like GSM8K and ProofWriter?

**Limitations:**

Yes.

---

> ### Author Rebuttal · Authors · 2024-08-07
>
> Thank you for your valuable feedback and questions. Below, we respond to the comments in ***Weaknesses (W)*** and ***Questions (Q)***.
>
> ---
>
> ***W1: Evaluation of the potential impacts of fine-tuning process on LLM's other abilities***
>
> In response to your suggestion, we conducted additional experiments using the LLaMA2-7b-base model to assess how fine-tuning on one specific task influences performance on other tasks. The results are presented in the table below:
> |Training|Test|||||||
> |  ----  | ----  | ----  | ----  | ----| ----  | ----| ---- |
> ||Bam.|2wiki.|hot.|Fever|Feverous|Vitaminc|SVAMP|
> |-|29.6|26.3|21.0|45.8|44.3|47.3|37.7|
> |Bamboogle|**32.0**|23.3|19.7|45.5|44.3|47.1|41.7|
> |Fever|28.8|25.4|22.3|**53.2**|47.0|49.3|40.3|
>
> In the table, '-' represents the base model without any fine-tuning. **Bold** indicates testing on the same task as the model was fine-tuned on.
>
> The results reveal that while fine-tuning on specific tasks often improves performance on the targeted task, it can lead to decreased performance on other tasks. To address this, we are considering strategies such as using a more diverse mixture of data for fine-tuning and incorporating additional regularization techniques. Interestingly, the improved reasoning ability may generalize to other domains (e.g., the improved performance of SVAMP dataset after fine-tuning).
>
> We will include discussions in the revised version. Thanks for your suggestions.
>
> ---
>
> ***Q1: LN35, what is the advantage of CPO comared to existing MCTS methods?***
>
> Our CPO utilizes computationally expensive ToT to construct preference data for optimization during the training phase, and can efficiently generate answers through greedy decoding during testing.
> ToT is a method that augment the reasoning capabilities of LLMs by using tree search algorithms (i.e., BFS and DFS) to guide multi-step reasoning. While existing methods also explore using MCTS (Monte Carlo Tree Search) to guide LLM reasoning, our choice of ToT offers several advantages:
>
> 1. Self-Evaluation: ToT leverages the LLM itself to evaluate its generated reasoning paths, eliminating the need for labeled data to train an additional value function or reward model, which MCTS requires.
> 2. Performance: According to our preliminary exploration, MCTS guided LLM decoding does not consistently outperform ToT, which is also observed by [14].
> 3. Efficiency: MCTS is generally more resource-intensive than the search algorithm used in ToT (i.e., BFS and DFS), especially in terms of computational resources and time, because it involves a large number of random simulations. Moreover, ToT, with its pruning mechanisms, offers a more efficient tree search process by reducing unnecessary explorations.
>
> ---
>
> ***Q2: LN117, considering online RL that regenerates the data using the updated LLM.***
>
> We chose not to employ online RL due to the significant computational overhead it incurs, as each update step involves sampling from the current policy model.
>
> Instead, we opted for an iterative approach as a compromise between online and offline settings. This method reduces computational demands while still allowing for periodic updates from the policy model. Details on our iterative setting can be found in Appendix C of our paper.
>
> ---
>
> ***Q3: Table1, CPO performance on GSM8K and ProofWriter.***
>
> Following your suggestions, we have included our results on GSM8K and ProofWriter in $\\textrm{\\color{blue}Table B}$ of the Rebuttal PDF. For the ProofWriter dataset, we sampled 300 instances for testing, as justified in our response to ***W1*** of **Reviewer VaVd**. For the GSM8k dataset, we evaluated the models using the full test set. As shown in the Table, CPO enhances performance by 1.5% on GSM8K and 1.1% on ProofWriter compared to CoT, demonstrating its effectiveness on these common reasoning tasks.

---

> > ### Comment · Reviewer_Nj7W · 2024-08-14
> >
> > Thanks the author for the update on the potential impacts of fine-tuning process, which confirms my concerns. I'd like to keep my current score.

---

> > > ### Author Response · Authors · 2024-08-14
> > >
> > > We appreciate your detailed comments and suggestions. In the revision, we will include the new results and the discussion on the potential impacts of fine-tuning process. Thank you!

---

### Official Review · Reviewer_VaVd · 2024-07-13

**Soundness:** 3
**Presentation:** 4
**Contribution:** 3
**Rating:** 6
**Confidence:** 4

**Summary:**

This paper presents Chain-of-preference optimization (CPO) a self-supervised learning extension of Tree of Thought (ToT). Rather than use ToT during test time, which takes exponentially longer than end-to-end sampling, this paper proposes to use ToT at training time to annotate data for DPO fine-tuning and then use the DPO-tuned model end-to-end at test time. Notably, the DPO annotations are performed by the untuned model itself being prompted to label inferences as either useful or not. This differs from other approaches that are trained on only full successful reasoning paths.

For certain QA and reasoning tasks, the paper presents experimental evidence that CPO yields stronger models than either (1) fine-tuning the same LM on positive examples (TS-SFT) as done in previous papers or (2) running regular chain-of-thought prompting.

**Strengths:**

1. The approach is an intuitive and straightforward improvement to multi-hop reasoning that that is much faster than tree-based inference procedures.
2. The approach is appealing as it doesn't rely on correct labels nor external LLMs other than the LLM being fine-tuned on its own preference data.
3. The methodology is well explained and easily reproducible.

**Weaknesses:**

1. While the authors run evaluation over several datasets, they only consider 300 questions per dataset which is quite small. A stronger evaluation would use the whole test sets.
2. Details about the baseline implementation are unclear. The authors refer to both "TS-SFT" and "TS-LLM" (l.206) which in the original paper refer to different things-- SFT refers to the model fine-tuned on training examples with reasoning traces pulled from both a gold-annotated dataset and/or reasoning traces from the model that led to correct answers. However, TS-LLM in the original paper refers to the result of an iterative refinement process.
3. The choice of datasets is somewhat odd. E.g. HotpotQA is meant to be evaluated against support documents, which might explain the very low scores. They also compare against approaches (ToT, TS-SFT) that were not evaluated on any of the QA or Fact Verification tasks under consideration.  This paper did _not_ consider the Game of 24 or GSM datasets, which both ToT and TS-SFT did evaluate.

Missing references:
1. Khalifa et al 2023: [GRACE: Discriminator-Guided Chain-of-Thought Reasoning](https://aclanthology.org/2023.findings-emnlp.1022.pdf)
2. Li et al 2023: [Learning Math Reasoning from Self-Sampled Correct and Partially-Correct Solutions](https://openreview.net/pdf?id=4D4TSJE6-K)

**Questions:**

* What metrics did you use for the QA datasets? many of them use more than exact match, e.g. HotpotQA uses F1.
* The discussion around ablations in 5.3 is unclear. Are the trends consistent across different models and datasets?
* It is odd to have only considered up to 200 instances for constructing preference pairs. Why is this experiment limited to such a small maximum? The difference between e.g. 160 and 200 seems much less important than the difference between 200 and 1000 or 1000 and 5000.
* How many total preference pairs do you end up training on? It would be helpful to include this number in the main body of the paper.
* How stable is this approach to different prompts for state evaluation? Did you experiment with other prompts?

**Limitations:**

yes

---

> ### Author Rebuttal · Authors · 2024-08-07
>
> Thank you for your valuable feedback and questions. Below, we respond to the comments in ***Weaknesses (W)*** and ***Questions (Q)***.
>
> ---
>
> ***W1: Evaluation using entire test sets***
>
> Thank you for your suggestion. We selected evaluation sets of 300 questions per dataset to manage the high computational demands of evaluating ToT (more than 230,000 A100 GPU hours across seven full test sets and three base models). Sampling a relatively small subset is consistent with common practices in the field; for instance, ToT by Yao et al. [46], Self ask by Press et al. [29], Verify-and-edit by Zhao et al. [50], and Contrastive CoT by Chia et al. [A] also utilize small evaluation sets or sampled subsets for similar reasons. We will release the subset we used to make our results reproducible.
>
> We also acknowledge that using small evaluation sets could introduce variance in the results. Following your suggestion, we have included evaluations on the entire test sets in $\\textrm{\\color{blue}Table A}$ of the Rebuttal PDF. Currently, we are only able to provide results for CPO and two baselines (i.e., CoT and TS-SFT), as the computational cost for evaluating ToT is substantial.
>
> Our findings is that the relative improvement of CPO over the baselines is consistent with our previous results. We will incorporate these expanded evaluations in our revised paper.
>
> ---
> ***W2: Details about the baseline implementation & terminology***
>
> We apologize for the confusion. Throughout our paper, TS-SFT refers to the model fine-tuned on training examples with reasoning paths discovered by ToT, as described in L176-178. Note that these reasoning paths do not necessarily lead to correct answers, as we do not assume access to ground truth in our setting. The term TS-LLM in L206 was a typo and will be corrected to TS-SFT.
>
> We will clarify these in the revision and consistently use the term TS-SFT to avoid misunderstandings. Thank you for your attention to detail.
>
> ---
> ***W3: Choice of datasets and comparison approaches***
>
> We understand the concerns about our selections and the comparability of our results with established benchmarks.
>
> 1. HotpotQA is typically evaluated against wikipedia pages. We selected this dataset because LLMs are generally pre-trained on Wikipedia [B]. This aligns with prior work that tests LLMs on HotpotQA by relying solely on the information encoded in the model's parameters like [C].
> 2. Our choice of tasks was driven by the performance of the models using the ToT method, which showed improvements on QA and Fact Verification.
> 3. Based on your suggestion, we have added performance for the LLaMa2-7B base model on the full GSM8k test set in $\\textrm{\\color{blue}Table B}$ of the Rebuttal PDF. As shown in the Table, on LLaMA2-7b-base, CPO enhances performance by 1.5% on GSM8K, compared to CoT. Regarding the Game24, we find ToT can only achieve less than 3% with LLaMA3-8b-base (also observed by [D]). Our method, which solely relies on the inherent capabilities of the LLM for self-improvement without assuming access to human annonated data like [14], faces significant challenges in improving upon these figures. In addition, while ToT was tested on Game24 using GPT-4 in the original paper, GPT-4 does not currently offer an interface that supports DPO finetuning, precluding our ability to apply our methods in that context.
>
> ---
> ***Q1: Metrics for QA datasets***
>
> We primarily reported accuracy as our metric (line 171) following [29]. Additionally, we added F1 scores for the three QA tasks in $\\textrm{\\color{blue}Table D}$ of the Rebuttal PDF. As shown in the table, the performance in terms of F1 scores is consistent with its corresponding accuracy.
>
> ---
> ***Q2: Ablations in 5.3: Are the trends consistent across different models and datasets?***
>
> Following your suggestion, we have included ablations and analysis across different models and datasets in $\\textrm{\\color{blue}Figure A, B, C(a) and C(b)}$ and  $\\textrm{\\color{blue}Table C}$ of the Rebuttal PDF. We find the trends are generally consistent across different models and datasets.
>
> ---
> ***Q3: Limitation of 200 instances for constructing preference pairs***
>
> We construct preference pairs with up to 200 instances for the following reasons:
>
> 1. In our experiments, approximately 200 samples (e.g., questions in the QA task) on average could generate about 6,531 preference pairs, suggesting that our CPO requires only a small number of samples by design.
> 2. Constructing preference data is a time-intensive process. The choice of 200 samples represents a practical trade-off between efficiency and effectiveness, allowing us to manage resources effectively while still achieving noticeable improvements.
>
> Following your suggestions, we added an analysis of performance trends from 0 to 3,000 samples in $\\textrm{\\color{blue}Figure C(c)}$ in the Rebuttal PDF. We observed that while performance increases with more samples, the preference data constructed from 200 samples already provides a significant boost in model performance.
>
> ---
> ***Q4: Total number of preference pairs used in training***
>
> On average, we trained on 6,531 preference pairs across all three LLMs and seven datasets. We appreciate your suggestion and will ensure to include this specific number in the main body of the paper for clarity and completeness.
>
> ---
> ***Q5: Stability of approach to different prompts for state evaluation***
>
> In our prior experiments, we tried different prompting templates for evaluation. We slightly changed the prompts by 1) adding more suggestive information, such as "a good annonator will classify it as"; and 2) replacing prompts with synonymous sentences. We find that the reward score returned by the LLM is robust to such kinds of changes. Balancing the performance and the length of different templates, we chose the current prompting template. We will include the discussion in the revision.

---

> > ### Comment · Reviewer_VaVd · 2024-08-12
> > **Thanks for the responses**
> >
> > Thank you to the authors for their careful rebuttal. I still have concerns about the evaluation given the authors have been selective about which results were shown in the rebuttal PDF.
> >
> > >  we have included evaluations on the entire test sets in Table A
> >
> > This is greatly appreciated, though this only includes 3 of the 7 considered datasets in the paper. Were the results on the other 4 datasets not positive findings? Running on those datasets is clearly within scope because you included SVAMP and Fever in Table C.
> >
> > > added performance for the LLaMa2-7B base model on the full GSM8k test set in Table B
> >
> > Thank you as well for this result. Table B is a good start but these numbers do not constitute a full results that could be added to the paper. Why is it missing the other baselines? I appreciate the improvements by about a point over CoT. However, TS-SFT (and ToT) consistently improve upon CoT. For the other datasets you were outperforming CoT by 3-7 points pretty consistently, but for GSM it is only 1 point-- are the TS-SFT and ToT baselines somewhere inside this 1 point spectrum (raising questions about statistical significance), or are they stronger than CPO?
> >
> > Moreover, others have found (e.g. in the [Gemma report](https://storage.googleapis.com/deepmind-media/gemma/gemma-report.pdf) that Llama-2-7B matches your 14.6% performance on GSM using just few-shot prompting. This suggests that CPO is not providing any benefits.
> >
> > > [Ablations]
> >
> > Thank you for these results, they have somewhat addressed my concern. Which datasets are the Figure C plots computed over?

---

> > > ### Author Response · Authors · 2024-08-13
> > > **Thank you for your feedback**
> > >
> > > Thank you for your detailed feedback. We appreciate your thorough review and would like to address each of your comments below.
> > >
> > > ---
> > >
> > > >Were the results on the other 4 datasets not positive findings?
> > >
> > > The full test sets for Fever, Feverous, and Vitaminc are indeed quite large, with approximately 10,000 instances each for Fever and Feverous, and over 50,000 for Vitaminc. Due to time constraints during the rebuttal period, we initially prioritized reporting performance within the QA domain. However, we have since conducted additional experiments, and we now present the results on all seven datasets:
> > >
> > > Model | | Bam | 2wiki | hotpotqa | fever | feverous |	vitaminc | svamp | Avg.|
> > > |  ----  | ----  | ----  | ----  | ----  | ----  | ----  | ----  |----  |----  |
> > > |LLaMa2-7b	|CoT	|29.6|	23.8|	20.7|	38.6|	51.2|	50.5|	37.7|36.0|
> > > |	|TS-SFT|	30.4|	24.1|	22.7|	40.3|	53.0|	53.0	|43.1|38.1|
> > > ||CPO|	**32.0**|	**26.1**|	**24.0**|	**45.1**|	**56.2**|**55.2**|	**46.0**|**40.7**|
> > > |LLaMa2-13b|	CoT|	48.0|	28.4|	27.0|	47.4|	49.9|	50.8|	40.3|41.7|
> > > |	|TS-SFT|	50.8|	29.0|	28.1|	48.0|	48.8|	53.8|	44.6|43.3|
> > > |	|CPO|	**52.0**|	**30.3**|	**28.5**|	**49.7**|	**50.7**|	**59.6**|	**50.0**|**45.8**|
> > > Mistral-7b|	CoT|	41.6|27.5|24.8|53.6|54.0|53.8	|65.3|45.8|
> > > ||TS-SFT|	41.6|30.3|25.1	|**56.0**|55.0|57.1|					59.0|46.3|
> > > |	|CPO|	**45.6**|**31.3**|**25.9**|55.7|**60.1**|**60.0**|						**69.3**|**49.7**|
> > >
> > > Our findings indicate that the relative improvement of CPO over the baselines is consistent with our previously reported results. We will clarify it and incorporate these expanded evaluations into the revised paper.
> > >
> > > ---
> > >
> > > > Why is Table B missing the other baselines?
> > >
> > > Due to time constraints during the rebuttal period, we prioritized demonstrating CPO's potential to improve performance on new tasks, including GSM8k, rather than emphasizing its superiority over other methods. However, as per your suggestion, we have now included the experimental results for ToT and TS-SFT on GSM8k. Since some settings need to be clarified before presenting the results, please refer to our response to the following comments for detailed results and conclusions.
> > >
> > > ---
> > >
> > > >Moreover, others have found (e.g. in the Gemma report that Llama-2-7B matches your 14.6% performance on GSM using just few-shot prompting. This suggests that CPO is not providing any benefits.
> > >
> > > The performance of 8-shot prompting on GSM8k using LLaMA-2-7B, as reported in the Gemma report, is comparable to our CPO using 4-shot prompting (see lines 169-170). We extended our evaluation to 8-shot prompting here, consistent with the Gemma report settings:
> > >
> > > ||ToT|CoT|TS-SFT|CPO|
> > > |  ----  |  ----  | ----  |  ----  |----  |
> > > |GSM8k|16.2|14.2|14.8|15.3|
> > >
> > > We found that our COT approach achieves similar performance to that reported in the Gemma report, with CPO further improving the results to 15.3%.
> > >
> > > ---
> > >
> > > > Are the TS-SFT and ToT worse than CPO (raising questions about statistical significance), or are they stronger than CPO?
> > >
> > > As shown in the table provided in our previous response, CPO outperforms TS-SFT, as confirmed by a bootstrap significance test, which yielded a p-value of 0.0011 (p < 0.05), indicating a statistically significant difference. However, ToT is superior to CPO. This result is expected, as ToT serves as the teacher method for our CPO method, guiding its improvements.
> > >
> > > ---
> > >
> > > >Which datasets are the Figure C plots computed over?
> > >
> > > For $\\textrm{\\color{blue}Figure C (a)}$ and $\\textrm{\\color{blue}(b)}$, as labeled on the x-axis, the results are reported for the Bamboogle and Fever datasets, respectively. For $\\textrm{\\color{blue}Figure C (c)}$, we present the results on the Bamboogle dataset. We will ensure clarity when we include these figures in the final revision. The reason for reporting results on only one dataset is the computational cost associated with using ToT for dataset construction (e.g., over 1740 A100 GPU hours for LLaMA 2-7B on the Fever dataset). However, we are continuing to conduct this ablation and will include the results in the revised version.
> > >
> > > ---
> > >
> > > We hope these efforts address your concerns. If you have any further feedback, we will do our best to respond.

---

> > > > ### Comment · Reviewer_VaVd · 2024-08-13
> > > > **Thanks for the expanded results**
> > > >
> > > > Thank you for including these additional results. They have significantly improved this portion of the submission, so I will raise my score to a 6.

---

> > > > > ### Author Response · Authors · 2024-08-13
> > > > > **Thank you for your support and raising the score**
> > > > >
> > > > > We greatly appreciate your prompt feedback and for raising the score. We will incorporate these additional experiments into the final revision and ensure clarity throughout the paper. Thank you again for your valuable insights and support!

---

> ### Author Response · Authors · 2024-08-07
>
> **References**
>
> [A] Chia et al. Contrastive Chain-of-Thought Prompting, arXiv:2407.03600
>
> [B] Shi et al. Detecting Pretraining Data from Large Language Models, ICLR 2024.
>
> [C] Wang et al. Causal-driven Large Language Models with Faithful Reasoning for Knowledge Question Answering, MM2024
>
> [D] Yang et al. Buffer of Thoughts: Thought-Augmented Reasoning with Large Language Models, arXiv:2406.04271
>
> [E] Zhou et al. LIMA: Less Is More for Alignment, arXiv:2305.11206

---

### Author Rebuttal · Authors · 2024-08-07

We thank all reviewers for their constructive feedback, and we have responded to each reviewer individually. We have also uploaded a Rebuttal PDF that includes:

- $\\textrm{\\color{blue}Figure A}$: Effect of the number of instances in generating paired thoughts;
- $\\textrm{\\color{blue}Figure B}$: Effect of dispreferred thoughts in optimization;
- $\\textrm{\\color{blue}Figure C}$: Component-wise Evaluations and Analysis;
- $\\textrm{\\color{blue}Table A}$: Evaluation using entire test sets;
- $\\textrm{\\color{blue}Table B}$: Experiment results: GSM8K on LLaMA2-7B; others on LLaMA3-8B;
- $\\textrm{\\color{blue}Table C}$: Sensitivity of data mixture;
- $\\textrm{\\color{blue}Table D}$: F1 scores on QA datasets;
- $\\textrm{\\color{blue}Table E}$: Illustrative examples of the reasoning paths preferred by CPO.

---

### Decision · Program_Chairs · 2024-09-25

**Decision:**

Accept (poster)

**Comment:**

This study focuses on solving complex reasoning problems using large language models. It introduces the Chain of Preference Optimization (CPO) paradigm to enhance the traditional Chain-of-Thought (CoT) method. The CPO approach uses the Direct Preference Optimization algorithm, a reinforcement learning method, to establish a preference order among paths in the Tree-of-Thought (ToT) and select the most preferred one for query answering. Comparative experiments show the effectiveness of CPO.

All the reviewers’ ratings for this paper are very close, ranging from “borderline accept” to “weak accept”. After the rebuttal, reviewers agree that the chain of preference optimization approach significantly improves the chain-of-thought without increasing the inference load. The experimental results clearly highlight this ability, and even indicate that CPO can sometimes outperform the unpruned ToT baseline. Furthermore, the paper is well-organized with clear notation and a smooth flow of concepts.

Although the paper's strengths outweigh its weaknesses, I highly recommend addressing all reviewers’ concerns and using the conclusions of the fruitful discussion during the rebuttal. Notably, the differences between CPO and DPO, together with the use of CPO instead of MCTS, should be emphasized in the paper. Additionally, the new experimental results and reasoning tasks should be incorporated in the paper or the appendix.